# Understanding the Health and Health-Related Social Needs of Youth Experiencing Homelessness: A Photovoice Study

**DOI:** 10.3390/ijerph19169799

**Published:** 2022-08-09

**Authors:** April Joy Damian, Delilah Ponce, Angel Ortiz-Siberon, Zeba Kokan, Ryan Curran, Brandon Azevedo, Melanie Gonzalez

**Affiliations:** 1Weitzman Institute, Community Health Center, Inc., 19 Grand Str., Middletown, CT 06457, USA; 2Department of Mental Health, Johns Hopkins Bloomberg School of Public Health, 624 N. Broadway, Baltimore, MD 21205, USA

**Keywords:** COVID-19, youth homelessness, health equity, social determinants of health, racial/ethnic minority health, LGBTQ+ health, photovoice, community-based participatory research, qualitative research

## Abstract

Purpose: Homelessness is a major public health problem facing millions of youths across the United States (U.S.), with lesbian gay, bisexual, transgender, questioning (LGBTQ+) youths and youths of color being disproportionately at higher risk. This study obtains an understanding of the health and health-related social needs of youths experiencing homelessness during the coronavirus disease (COVID-19) pandemic. Methods: A total of 14 youths between the ages of 14 and 24 who (a) lived, worked, or attended school in New Britain, Connecticut (CT) and (b) had at least one experience of homelessness or housing insecurity worked with the research team to conduct a needs assessment regarding youth homelessness. Using photovoice, a community-based participatory research method, participants created photo narratives to share their stories and recommendations for community change. The main goals of photovoice are to enable participants to (1) record and represent their everyday realities; (2) promote critical dialogue and knowledge about person and community strengths and concerns; and (3) reach policy makers. Results: Most of the participants identified as youths of color, and half of the participants identified as members of the LGBTQ+ community. Three major themes that appeared in the youths’ narratives include the following: mental health and substance use challenges, trouble accessing basic human needs, and lack of a social support system. Conclusion: This study uplifts and empowers a vulnerable population to increase visibility around a major public health challenge from their own lived experiences. Despite the challenges that were voiced, many participants shared a sense of hope and resiliency. The major themes endorsed by the youths has the potential of informing practitioners and policy makers of how to better address the needs of youths experiencing homelessness, particularly those most at risk.

## 1. Introduction

Homelessness, broadly defined as the experience of sleeping in places in which people are not meant to live, staying in shelters, or temporarily staying with others (e.g., couch surfing) and not having a safe and stable alternative [1], is a major public health challenge facing 4.2 million youths across the United States (U.S.) [2]. The most recent national reports of homelessness show that 1 in 10 young adults aged 18–25, and at least 1 in 30 adolescents aged 13–17, experience some form of homelessness unaccompanied by a parent or guardian over the course of a year, with the caveat that these national estimates often underreport the actual prevalence of youth homelessness, given the methodological challenges in assessing this issue [2].

National prevalence and incidence reports show significant disparities in experiences of housing insecurity, with lesbian gay, bisexual, transgender, questioning (LGBTQ+) youths and youths of color being more likely to have unstable living conditions, compared to their respective heterosexual and white counterparts [1,2]. The LGBTQ+ collective as a whole has been marginalized, due to a myriad of longstanding factors intersecting on political, social, legal, cultural, and religious grounds. The origin of LGBTQ+ oppression has been attributed to repressive constructs, including patriarchal norms, homophobia, and sexual repression tactics as a means of control [3] (p. 26). The refusal of the rights of LGBTQ+ people is often used under the guise of combating “gender ideology” often found in governments with exclusionary ideologies [4] (pp. 1459–1480). Similarly, the enforcement of “racial ideology”, through racialized social systems and societal norms, contributes to the longstanding health and social disparities that racial and ethnic minority populations experience [5] (pp. 63–82). These paradigms are codified into legislation and socio-cultural norms that infringe on the right of LGBTQ+ communities and people of color to have access to appropriate resources and opportunities to lead healthy lives.

There is a significant amount of literature that describes the high prevalence of mental, behavioral, and physical health issues among youths experiencing homelessness, including high rates of alcohol and substance use [6] (pp. 994–1009), [7] (pp. 533–543), [8] (pp. 773–782); polytrauma and abuse [9] (p. 16), [10] (p. 216), [11] (pp. 894–910), [12] (pp. 191–198); sexually transmitted diseases [13] (p. 15), and advanced chronic respiratory illnesses [14] (354–375). Youths experiencing homelessness also face a variety of challenges associated with health-related social needs, including food insecurity, limited social support, poor academic success, and a lack of health insurance and access to quality health services, all of which contribute to mental health illnesses and exacerbate chronic health conditions [15] (pp. 461–462), [16] (p. 245), [17] (pp. 653–657), [18] (pp. 372–375). An emerging amount of literature reports how the coronavirus disease (COVID-19) pandemic has contributed to an exacerbation of these pre-pandemic health and health-related social challenges, with LGBTQ+ youths and youths of color being disproportionately impacted [9] (p. 16), [11] (pp. 894–910), [17] (pp. 653–657).

Both qualitative and quantitative methods have been used to identify the needs of homeless youths. Two of the studies that we examined used semi-structural interviews with participants [9] (p. 16), [17] (pp. 653–657), while others used self-reporting survey methods [19] (pp. 381–382). Semi-structured interviews may lead to social desirability bias from the participants by answering according to what they believe the interviewer wants to hear, whereas the studies that used self-reporting surveys or online messaging could encourage selection bias, since responses from homeless youths who did not have access to technology were excluded [9] (p. 16), [11] (pp. 894–910), [17] (pp. 653–657), [19] (pp. 381–382). Such methodological limitations suggest that additional research is needed.

Photovoice is a community-based participatory method that was developed by Wang et al. as a way to conduct community needs assessments in partnership with community members whose voices and perspectives are typically excluded from public discourse and community decision-making [20] (pp. 369–387). As a research approach that is theoretically grounded in critical social theory and Paolo Freire’s work on the pedagogy of the oppressed, photovoice provides fertile ground for researchers to work in partnership with community members to not only identify community needs, but to also critically examine the root causes from which those needs developed. This process of critical examination and analysis of social issues provides opportunities for researchers and community members to collaboratively develop recommendations for social interventions and policies that leverage both academic and lived expertise. This method equips participants with cameras to capture their everyday experiences and construct accompanying narratives that generate knowledge about health issues of concern [21] (pp. 310–325). The main goals of photovoice, which guided the objectives of this study, are to enable participants to (1) record and represent their lived experiences; (2) promote critical dialogue and knowledge about person and community strengths and concerns; and (3) reach policy makers [22] (pp. 185–192).

Additionally, given the significant influence of social systems on the marginalization of LGBTQ+ communities and people of color, it is important for research to utilize a framework that extends beyond a study of individual characteristics, and includes an analysis of influences stemming from systems that individuals are part of. The socioecological model has been applied in various disciplines to examine the interconnected web of factors associated with human behaviors and outcomes, and offers a framework to encourage an openly critical examination of the societal and structural injustices that contribute to experiences such as youth homelessness [23] (pp. 282–309), [24] (pp. 1479–14950), [25] (pp. 469–285). 

In this study, we applied photovoice and the socioecological model to obtain a nuanced understanding of the individual, familial, community, and societal factors that contributed to the health and health-related social needs of youths experiencing homelessness. The relevance and timeliness of this research question was derived from a priori data collection with adult professionals currently working with youths experiencing homelessness, which is discussed in greater detail in a separate paper [26]. The research team believed it was important that the current study was guided by, at the very least, voices of those who work closely with individuals and communities directly impacted by youth homelessness. This photovoice project provided an opportunity for young people with experiences of homelessness to (1) participate in training aimed at improving their knowledge and skills in photography and advocacy, (2) construct photo narratives, sharing their perspectives about the factors that contribute to youth homelessness, and (3) develop and disseminate recommendations to address the factors that contribute to youth homelessness. To our knowledge, this is the first study using photovoice to examine these issues during COVID-19, and that intentionally targets youth populations at the greatest risk for homelessness (e.g., LGBTQ+ youth, youth of color). 

## 2. Methods

### 2.1. Participants: Eligibility, Recruitment, and Human Subject Protection 

Eligible participants were people between the ages of 14 and 24 who, at the time of enrollment, (a) lived, worked, or attended school in New Britain, Connecticut and (b) had at least one experience of homelessness or housing insecurity. The research team sought the guidance of advisors who have each been working with marginalized populations in New Britain for over three decades. When the authors expressed an intentional focus on engaging LGBTQ+ youths and youths of color in the study, the advisors proceeded to refer the research team to other community leaders and community-based organizations that were known to serve these particular populations. The research team scheduled one-on-one meetings with community stakeholders who served young people in New Britain, including young people experiencing homelessness, to discuss the aims of the project. The authors established rapport with these leaders and organizations, who, in turn, provided recommendations for additional entities to contact to add further support to the recruitment of LGBTQ+ youths and youths of color. Recruitment occurred primarily through referrals from these community stakeholders. The research team proceeded to administer consent forms and assent forms either in-person or through Zoom to eligible youths interested in participating in the study. The study protocol was reviewed and approved by the Community Health Center, Inc. Internal Review Board prior to recruitment.

### 2.2. Study Procedures 

Resources: The research team was comprised of a senior researcher, two research associates, a policy fellow, and a multimedia specialist. The research team was supported by senior clinical and community advisors who contributed to previous photovoice efforts in New Britain focused on health promotion efforts among underserved youths. The research team used the Rutgers Photovoice Training Manual [27] to construct a photovoice curriculum comprised of the following four components: (1) photovoice trainings, (2) photo-taking fieldwork, (3) one-on-one meetings, and (4) group sharing and analysis, each of which are described in greater detail below. 

Photovoice Training: During February 2021, two separate groups convened concurrently, either on Zoom or in-person. During March 2021, one group convened on Zoom. Researchers facilitated 4 weekly training sessions, each lasting 2–3 h, so that participants could understand the purpose of the project, develop photography skills, learn about the theoretical underpinnings of photovoice and the socioecological model, and draft ideas to answer the research question through their photography. Each training session began by reviewing the ground rules and completing an icebreaker to build team unity and cohesion. Disposable cameras were provided to 2 of the 14 participants who indicated that they did not have a phone that allowed them to take photos.

The first training session was focused on relationship building and providing an overview of the project. The second and third training sessions, facilitated by the multimedia specialist, focused on teaching participants the best practices for photo taking, as outlined in the Rutgers Photovoice Training Manual. Additionally, participants were given opportunities to practice the photo-taking techniques during these training sessions and receive feedback from the facilitators. The fourth and final training session focused on supporting participants for their fieldwork in capturing, through images and narratives, the health and health-related social needs of youths experiencing homelessness based on their own lived experiences. To do so, the research team supported participants in developing lists of factors that contribute to homelessness at the levels illustrated through the socioecological model. After the participants developed their list of factors, they were trained to engage in their photo-taking fieldwork. The research team informed the participants of the importance of exercising caution when taking photos, and to refrain from taking photos that might jeopardize their psychological and/or physical safety (e.g., capturing photos of illegal activity). A handout on photo-taking ethics and guidelines that the research team created was reviewed with the participants, which included a script to request permission to take photos and the contact information of the research team. Participants were instructed to carry the handout with them while taking photos.

Photo-taking Fieldwork and One-on-One Meeting: Participants were asked to take at least 25 photographs on factors that contributed to youth homelessness in New Britain over the course of 2–3 weeks. Participants subsequently reached out to the research team to schedule a one-on-one meeting, where they were asked to go through each of the photos they took, briefly explain what the photograph represented for them, and select their 5 favorite photos that they wanted to use for their photo narrative. After the photos were selected, a member of the research team worked with the participant to construct the narrative using the following three questions: (1) what is in the picture?; (2) what does this picture represent and how does it answer the research question?; (3) what recommendations do you have on how to address this issue? After the photo sharing and one-on-one meeting was complete, participants submitted their top 5 photos and narratives to the research team, with the understanding that their submission would be shared publicly, including with other participants.

There were slight modifications to the one-on-one meetings convened in person in February 2021 and on Zoom in March 2021. Methods of the in-person February 2021 group process included all of the same elements outlined above, with a few minor modifications. While the participants shared their responses about the 5 selected photos, the researcher typed their responses verbatim. After the sharing was complete, the researcher and participant reviewed the narrative and edited it together, then the participant created a title for the photo. If the participants were not able to work with the researcher to get through all of the photos during the allocated time they had together, the participant was asked to complete the activity on their own, email or call the researcher for support if needed, and send the photo narratives to the researcher once they were complete. Methods of the Zoom March 2021 group process included all of the same aforementioned elements, with the addition of the one-on-one sessions being recorded with permission from the participants, so that the conversations could be transcribed.

Group Sharing and Analysis: Researchers facilitated a final recorded meeting conducted either in person or on Zoom, where each participant had 5–10 min to share their photos and narratives with their group. Space was provided to allow participants to respond to each other’s photovoice stories, and construct initial collective themes for the photos. After participants completed all four components of the project, they received a stipend of USD 500 for 20 h of work. 

The selection of photos and identification and discussion of themes by the participants was a key component of the analysis, and ensured that the participatory process of the study was preserved. One member of the research team developed an initial codebook of potential themes, and collated the various sources of data, including the following: (1) notes taken by members of the research team during the sessions; (2) final products created by the participants, inclusive of photos and narratives, and (3) recordings of the sessions, including the final meeting. Both the codebook and collated data were then shared with two additional research team members, who independently reviewed and coded the data. Inter-rater reliability was determined by cross-checking analyses of different segments of the texts to ensure congruency, discussion of discrepancies was facilitated by the most senior member of the research team, and consensus was achieved.

### 2.3. Dissemination: Youth Advocacy Training and Multimedia Project 

The second part consisted of a two and a half month multimedia advocacy project, aimed at giving participants the opportunity to create multimedia products as a means of disseminating information about the issues identified during the photovoice project. Of the 14 participants who completed the photovoice project, 7 completed the multimedia advocacy project. For this project, participants attended 6 weekly 2-h training sessions to teach them about policy and advocacy, stakeholder analysis, power building, gathering evidence and doing due diligence, and producing a podcast. After training, participants had 4 weeks to work on their multimedia projects, supported by research staff through weekly one-on-one meetings. After participants completed all of the project components, they received an additional stipend of USD 500 for 20 h of work.

The photovoice photo narratives, multimedia pieces, podcast, and summary documents (e.g., policy briefs, manuscripts, presentations, etc.) were made publicly available on the Weitzman Institute website, which was subsequently disseminated to community members and policy makers in New Britain.

## 3. Results

### 3.1. Photovoice Participants

A total of 11 youths participated in the virtual sessions and 3 youths participated in the in-person session. Reasons for not participating in the training sessions included conflicts with their work schedule, disruptions resulting from the experience of homelessness, such as not having a space to attend sessions with WIFI access, and family commitments. Among the 14 youth participants, the majority of participants identified as being a person of color (*n* = 12) and female (*n* = 8). The mean age was 19.1 years. Half of the participants self-identified as part of the LGBTQ+ community (*n* = 7); two participants preferred not to provide information regarding their sexual orientation. The most common types of housing insecurity experienced by youth participants was living in a place not meant for human habitation (e.g., car, streets, etc.; *n* = 8) and emergency shelter (*n* = 8), although it is worth noting that the majority of participants (*n* = 11) reported experiencing more than one form of housing insecurity. 

### 3.2. Qualitative Findings

A total of 70 photos and corresponding narratives (5 from each participant) were presented during the group discussions. After an iterative review, coding, and analysis of the photos, narratives, and group discussions, the following three multi-faceted and inter-related themes emerged: (1) mental health and substance use challenges; (2) basic human needs, and (3) social support system. 

Theme 1: Mental Health and Substance Use Challenges: Participants described a variety of concerns related to mental health and substance use, such as depression, anxiety, trauma, alcohol and illicit drug use, and loneliness. A major factor associated with mental health challenges that participants endorsed was exposure to domestic and community violence. One participant expressed a feeling of “hopelessness” due to having a male family member that was repeatedly abusive and becoming “really depressed since it was really hard to go through the abuse to feel that no one could help me” (Figure 1).

Another participant described being verbally abused by her father after she was diagnosed with attention deficit/hyperactivity disorder (ADHD) and anxiety. At the community level, daily exposure to gun violence, bullying, and general lack of personal safety was a common theme described as triggering post-traumatic disorder (PTSD), depression, and anxiety. One participant described the use of weapons by law enforcement as a means of clearing out homeless encampments, as shown in the following statement:

The police threw out all the homeless and tore down the tent cities. The police came in with machetes and tore the tents while we slept in them. Families and elderly and the otherwise alone in this world lived there, myself included. There were injuries when that happened as you would expect when someone thrusts a blade in the tent you’re sleeping in. There needs to be set places where the homeless can camp until they can afford to get housing (Figure 2).

Participants also described how the experience of housing instability itself contributed to their mental and behavioral health challenges. A third of the participants described the pervasiveness of alcohol and illicit drug use in the homeless community, and some shared how they too turned to these substances to cope with their situation. One participant described the vicious cycle of alcohol and substance use further contributing to her state of housing insecurity and not being able to purchase basic necessities, as the limited amount of funds she had was used towards purchasing alcohol and drugs. In addition, a few participants noted how the pandemic and transition to virtual learning took a toll on their mental health. One participant with diagnosed ADHD noted having a “horrible experience with virtual school” (Figure 3) and becoming more distracted during online learning, which subsequently further contributed to feeling anxious and depressed.

Nonetheless, the majority of the participants endorsed an openness and need for more professional mental health and substance use services. Several participants noted the need for more mental health classes and alcohol and substance use treatment programs in schools and communities targeted towards persons experiencing housing instability. Participants described the need for greater affordability and accessibility of therapy, and for referring entities to close the loop on their referrals to mental/behavioral health services, since a referral does not guarantee that the participants were able to successfully secure the needed therapeutic services. As one participant said, “It’s not just housing first. If my mental health isn’t right, I’ll experience homelessness again. I was referred for mental health services, but the referring organization did not have a record of this, so I couldn’t get a telehealth visit” (Figure 4). A couple of participants also described societal challenges of criminalizing persons with alcohol/substance use problems. One participant noted, “People with drug problems shouldn’t be in jail. They need to be in a rehab facility. Staff should be reaching out to them and try to get them case management.”

Theme 2: Basic Human Needs: Participants provided vivid photos and narratives that illustrated the substandard living conditions that come with housing insecurity. Participants noted inadequate heating and lighting, poor bedding, and general concerns about safety in shelters and when sleeping in the streets. All participants also endorsed significant challenges accessing other basic human needs, including food, clothing, employment, and transportation. A couple of participants noted challenges with the Supplemental Nutrition Assistance Program (SNAP). One participant explained, “When you are homeless you don’t have a stove or even a campfire to cook your food. With food stamps you are not able to buy hot food, it’s mainly cold food. You are paying out of pocket for this because you can’t make your own food to eat” (Figure 5). Similarly, another participant noted experiencing food insecurity due to minimal benefits from food stamps, as shown in the following statement: “There are times when food stamps gives you a terrible amount of money. When I was in college, they gave me $15 a month. What can I buy for $15 a month?” (Figure 6).

Participants also described the high expense of doing laundry, and sometimes having to choose between having clean clothes, or trying to feed themselves. Similarly, participants noted challenges accessing reliable transportation due to limited bus routes or being dependent on family members for transportation. For example, one participant gave the following statement:

I would get out really late and the bus would stop running. I got off Saturday, Sunday buses wouldn’t work in New Haven, so I would sleep in train station. It was a public space and more secure and I needed it to work. I wasn’t thinking about safety, I needed money because if I don’t make money I will always be couch surfing. (Figure 7).

Both clothing and transportation challenges were cited as barriers related to overcoming financial hardship. One participant noted how not having professional clothing made it difficult to be presentable during job interviews, thereby making it difficult to secure gainful employment, as shown by the following statement: “Many of the clothes would not be appropriate for job interviews. I’m not looking for anything extreme—just a nice button down shirt for an interview” (Figure 8). Similarly, another participant noted the following: “A lot of the money I make goes towards bus fares. I pay 15.75 to get to and from work per week. It’s hard to pay for rent and money for transportation” (Figure 9). Participants also noted supplies that have become essential during the current COVID-19 pandemic, including personal protective equipment (PPE) and resources for online learning, such as a laptop and proper desk/space to study.

Theme 3: Social Support System: Participants noted challenges with having a proper social support system in place, and the need for greater government engagement. Several participants described being ostracized and ending up on the streets after disclosing their sexuality or gender identify to their family members. For example, one participant noted the following:

My mom disagreed with me being a trans man. When she kicked me out, it made me know I shouldn’t be afraid of being who I am. She would made me feel less worthy of being who I was. I would feel depressed for not being able to embrace me (Figure 10).

Participants provided several recommendations for how policy changes can better support youths experiencing housing insecurity. One recommendation reiterated by several participants was the need for greater investment in preventive services that target youths at risk of becoming homeless, so that this population can avoid housing insecurity and the related health and social challenges that come with this state. Preventive services mentioned include resources that promote healthy parenting skills that help families understand how to support LGBTQ+ youths; professional development and career exploration options for youths before they leave school, and strengthening and expanding after-school programs and other opportunities for youths to be immersed in safe, healthy spaces in the community.

Conversely, participants noted several community-based protective strategies that supported their mental and emotional well-being, and boosted their resilience. A few participants described the positive effect of having access to books, stuffed animals, and art supplies through school and shelters as a healthy outlet for coping. Others noted the impact of having positive adults at school and in the community. One participant said, “I remember going to the Boys and Girls Club as a boy and learning to play basketball there. I made friends with every staff member. They supported me when I was going through bullying” (Figure 11).

Despite the challenges the homeless youth participants faced, almost half of the participants still had hope and shared positive coping mechanisms. Lazarus and Folkman (1984), one of the pioneers of coping theory, define coping as “constantly changing cognitive and behavioral efforts to manage specific external and internal demands that are appraised as taxing or exceeding the resources of the person” [28] (p. 141). Other scholars have built on the work of Lazarus and Folkman by examining specific examples of coping theory in practice, including, but not limited to, *positive (adaptive) coping*, in which one uses healthy cognitive and behavioral strategies to deal with a given stressor, as well as *maladaptive coping*, which can be harmful and exacerbate mental health challenges [29] (pp. 459–466). In this study, we recognize how several participants are able to endorse the former in dealing with the trauma and challenges of experiencing homelessness. One participant who used a photo of dark figures and a colored sky stated, “If you look at it you see that towards the back there is more brightness but towards the front there is more darkness. I use it to describe life in a way. You go through darkness and storms to get the final outcome.” (Figure 12) Another encouraging life perspective stated from a participant was, “I didn’t want to see the positive and only wanted to stay in the negative. This made me think that everything negative has a positive and everything positive has a negative. You just got to give it time.” (Figure 13) The ability to capture moments through photovoice gave the participants a time of reflection and empowerment, shown by the following statement: “I took these picture because I wanted to remember my time here I am still positive and haven’t lost hope.” (Figure 14)

## 4. Discussion

This study provides insight into the multiple levels of factors that contribute to youth homelessness in New Britain, CT, while also providing recommendations that community stakeholders can take into consideration as they work to eradicate youth homelessness. The content that was shared by participants highlights the need for community providers, policymakers, and leaders to make changes to the ways that they provide services, resources, and support to young people experiencing homelessness. The youth participants noted the following three key areas that require urgent attention from community change makers: (1) developing and implementing more effective strategies to address mental health and substance use, specifically interventions that are trauma-informed and foster a collaborative approach to treatment, in which a provider works in partnership with a client to work towards healing and recovery, (2) providing better resources and wrap-around services to help satisfy basic human needs, and (3) implementing more preventative services intended to build healthy family units, resulting in safe and secure homes and strong social support systems that prevent the occurrence of youth homelessness.

The strengths of this study center on uplifting and centering young people’s voices. Amplifying young people’s voices creates an opportunity for the key findings to reflect the lived experiences of those who are often overlooked when discussing and making decisions about youth homelessness. By involving youths in this study, it empowers the participants to be engaged in an issue directly affecting their lives, and to use the knowledge and tools learned to continue advocating for change beyond the formal study period. Since in-person programming was focused on shelter residents, the additional virtual format drew from a broader pool of participants that experienced housing insecurity, including those who were outside the shelter system. This allowed this study to highlight the diverse and nuanced pathways into housing insecurity and shed light on the prevalence of homelessness and housing insecurity outside of the homeless service system.

There are several limitations to this study. First, the COVID-19 pandemic resulted in multiple instances of rescheduling trainings and meetings due to related illnesses among both participants and research team members, which presented challenges around engagement and ensuring high-quality training and learning. Additionally, the study sample size may not be large enough to make the results generalizable and applicable to all youths experiencing housing insecurity. The same can be said regarding the selection of participants, who lived in the same geographic region and whose race, sexual orientation, and gender may not necessarily be representative of the state of Connecticut or the U.S. Another limitation of the results of this study is that photographs may not fully capture the complex and nuanced challenges of housing insecurity, as participants may not have been able to, or felt comfortable with, documenting certain aspects of their lived experiences. Lastly, while this study was participatory by design, and intentionally focused on marginalized populations disproportionately impacted by youth homelessness, specifically LGBTQ+ youths and youths of color, the study did not incorporate theories including intersectionality, critical race theory, and queer theory, all of which have profoundly shaped today’s scholarship by calling attention to how historical and present-day injustices contribute to the ongoing challenges experienced by marginalized communities. Thus, future research that further explores youth homelessness grounded in the aforementioned theories is warranted and would provide a more nuanced understanding of how overlapping identities influence and are associated with this public health challenge. 

## 5. Conclusions

The results from this study provide valuable insights for practitioners and policymakers. Many photos and narratives highlight problems that are a result of policy decisions and can be ameliorated by developing interventions that cut across traditional service silos of health, housing, and employment. For instance, as participants cited clothing and transportation barriers to seeking gainful employment that would sustain a path out of homelessness, housing interventions should consider how to enhance access to factors that lead to successful job attainment. Similarly, given the impact of housing status on physical and mental health, housing interventions should consider the potential to impact health outcomes. This aligns with a growing body of literature that explores linkages across policy domains and service silos [30] (pp. 2374–2377), [31] (p. 6).

This study is also of great methodological relevance. This study highlights the need to involve youths in future research, especially on topics that directly impact them. The use of photovoice created an environment that allowed youth participants to share their lived experiences, learn from peers, and actively contribute to the identification of common themes among the participant narratives. An environment of trust, where active communication flourished, was built over time, through numerous weeks of work with the research team. Photovoice encouraged participants to discuss policy implications that stemmed from their personal experiences and disseminate lessons learned in the community. The value of this approach to research should be considered for future studies that document lived experiences of youths and other vulnerable populations. In addition, virtual arrangements enhanced the pool of participants and accessibility of the study, as some people experiencing housing insecurity were unable to attend in person sessions. Lastly, research aimed at understanding the experiences of homeless youths during a time of global crisis can help inform and create recommendations to ensure better supports are in place, and to prevent reoccurrences of the same challenges in the future.

## Figures and Tables

**Figure 1 ijerph-19-09799-f001:**
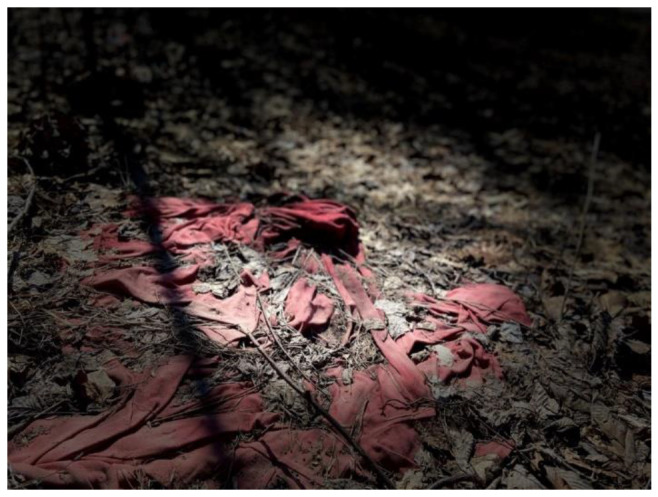
Sense of hopelessness and depression after experiencing repeated acts of domestic violence.

**Figure 2 ijerph-19-09799-f002:**
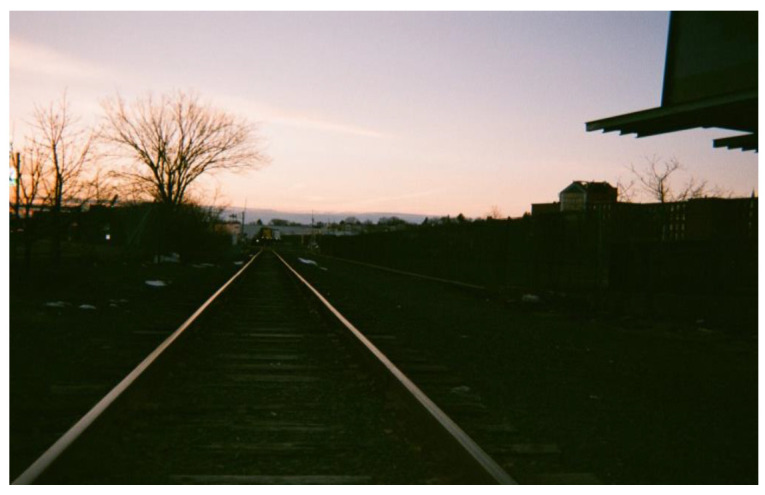
PTSD associated with violent demolition of tent cities.

**Figure 3 ijerph-19-09799-f003:**
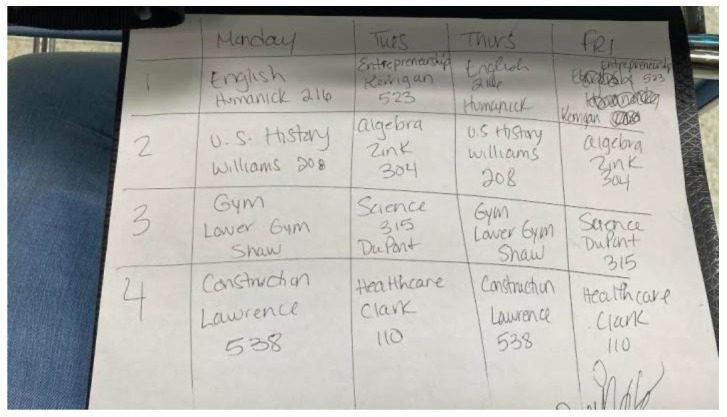
Dealing with the challenges of virtual learning for a participant with ADHD.

**Figure 4 ijerph-19-09799-f004:**
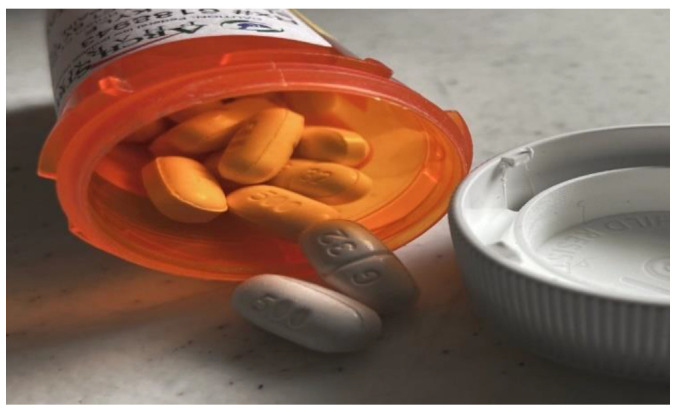
Challenges with inappropriate use of drugs/self-medicating.

**Figure 5 ijerph-19-09799-f005:**
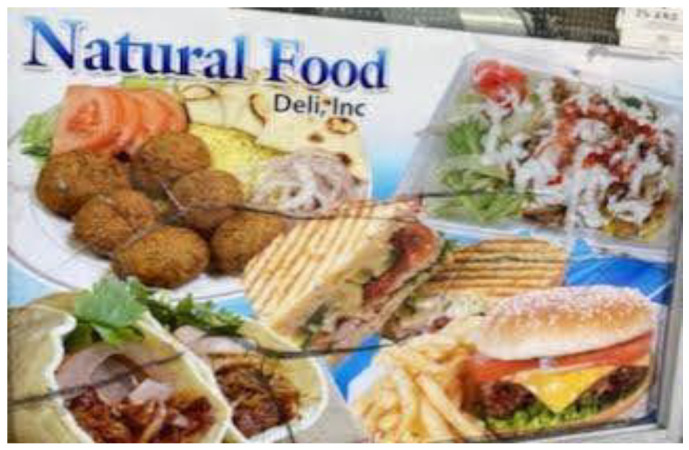
Dependency on pre-cooked meals.

**Figure 6 ijerph-19-09799-f006:**
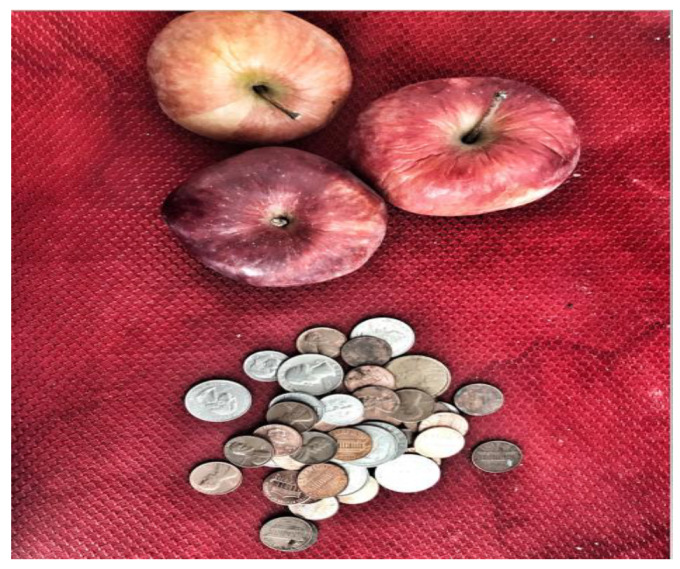
Food insecurity.

**Figure 7 ijerph-19-09799-f007:**
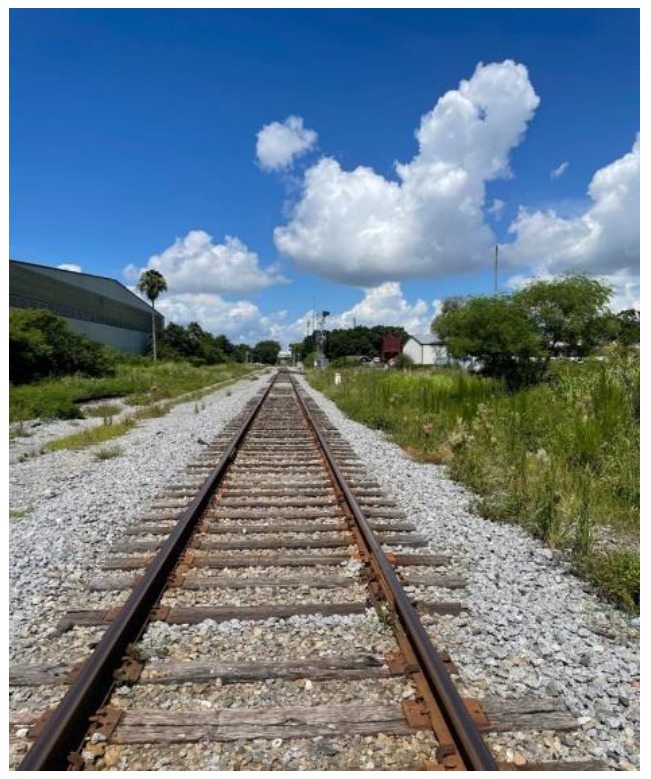
Challenges accessing reliable public transportation.

**Figure 8 ijerph-19-09799-f008:**
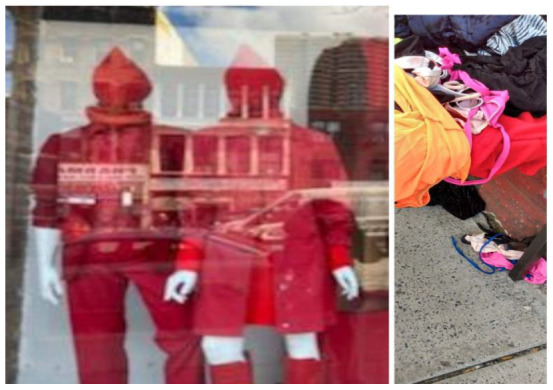
Lack of access to professional attire.

**Figure 9 ijerph-19-09799-f009:**
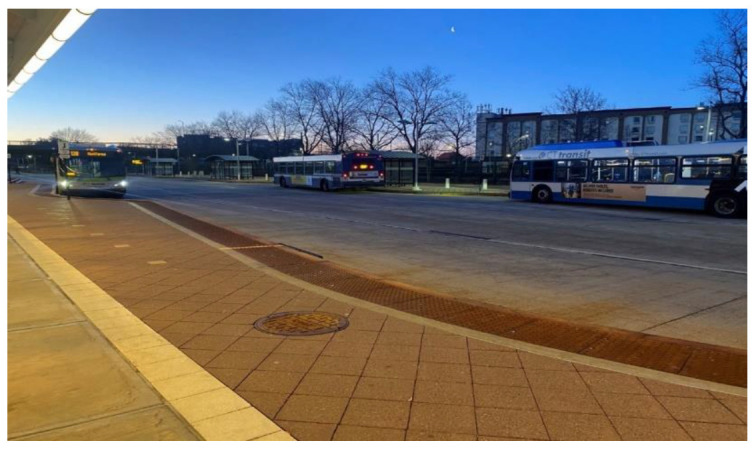
High-cost public transportation.

**Figure 10 ijerph-19-09799-f010:**
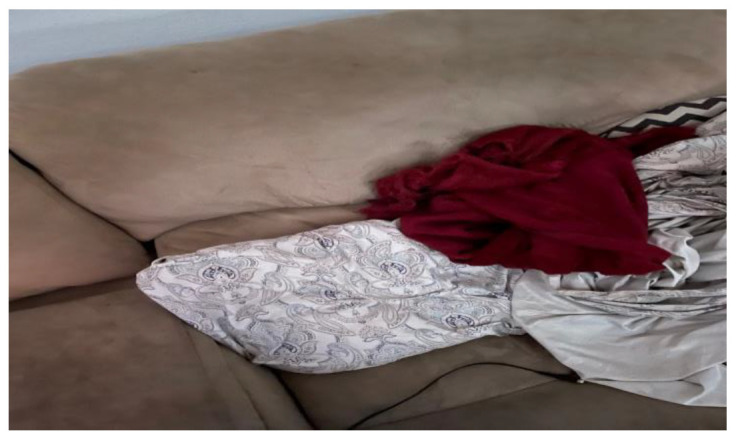
Couch surfing after family disownment.

**Figure 11 ijerph-19-09799-f011:**
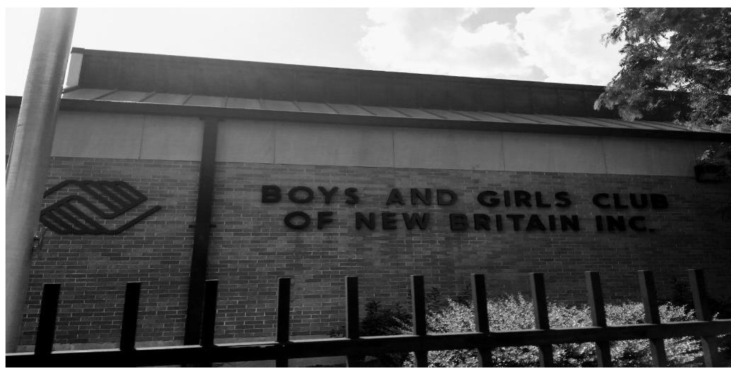
Youth-serving community organizations as places of refuge and support.

**Figure 12 ijerph-19-09799-f012:**
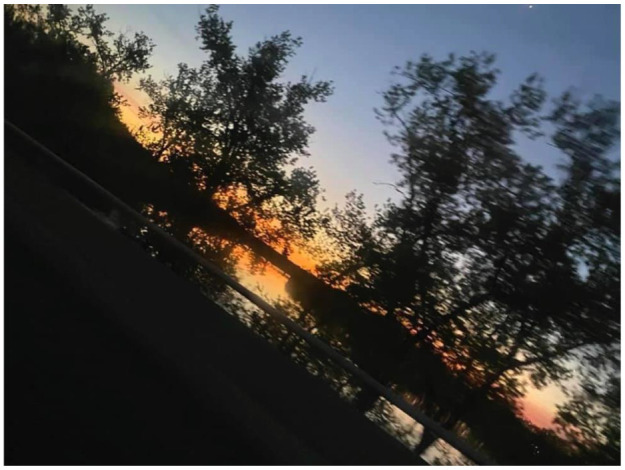
An encouraging view on the participant’s way to work.

**Figure 13 ijerph-19-09799-f013:**
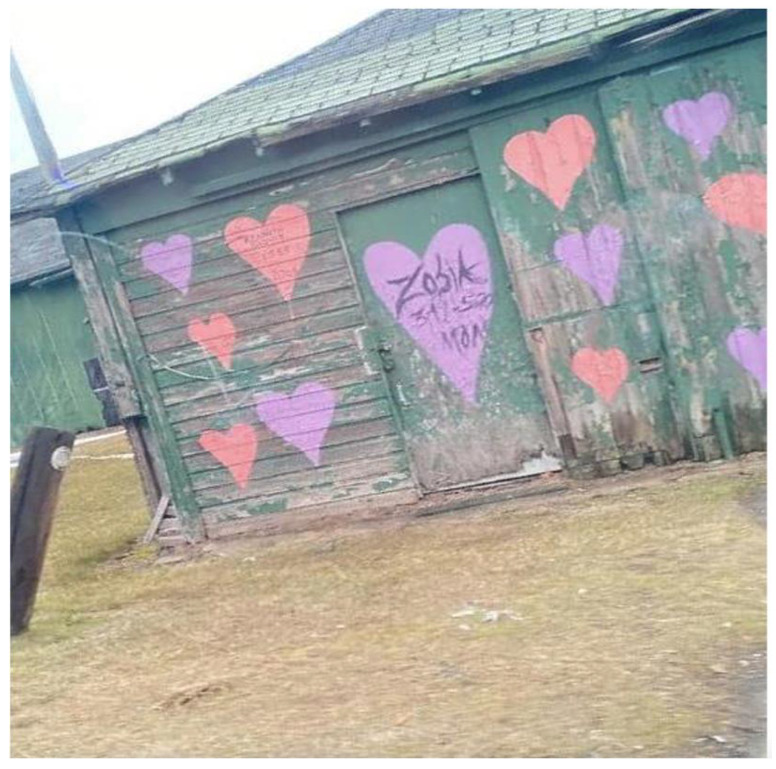
A rundown building with colored hearts representing optimism in the face of challenges.

**Figure 14 ijerph-19-09799-f014:**
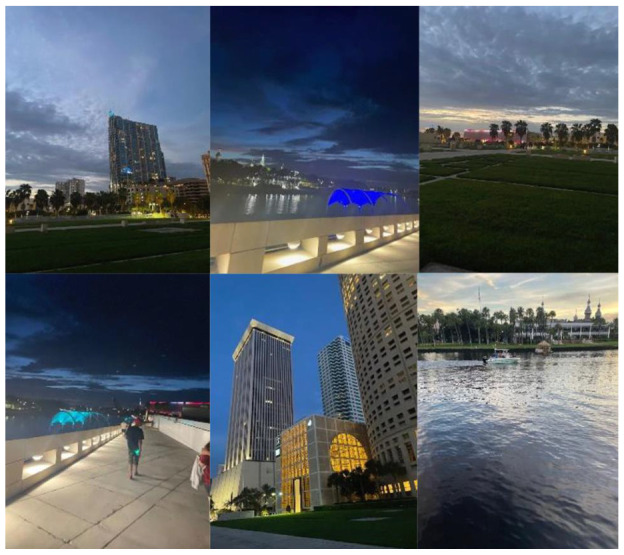
Images that represent remembering a time with a family member that brings positivity and hope.

## Data Availability

The data that support the reported results are available upon request to the corresponding author.

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
