# Peer review of "Understanding the Health and Health-Related Social Needs of Youth Experiencing Homelessness: A Photovoice Study"

_ijerph, 2022, doi:10.3390/ijerph19169799_

Round 1

Reviewer 1 Report

The revised version is significantly improved. The presentation is much better now. I hope I can see more pictures and quotes, but the current version is also fine. 

Author Response

The authors thank the editor for this comment, and overall, for both the editor and reviewers’ comments that helped the authors think more critically about the study and thereby, improve the manuscript.

Reviewer 2 Report

I agree with the revision that you have done. Now is ready to be published

Author Response

The authors thank the editor for this comment, and overall, for both the editor and reviewers’ comments that helped the authors think more critically about the study and thereby, improve the manuscript.

This manuscript is a resubmission of an earlier submission. The following is a list of the peer review reports and author responses from that submission.

Round 1

Reviewer 1 Report

This qualitative study adopted a photovoice approach to explore the homeless issue among minority youth. The purposes and findings significantly contribute to the current literature on such topics. I have the following suggestions for the authors.

1. Line 98-99. Is “understanding of the health and health related social needs of youth experiencing homelessness” the research question the research team gave participants? If yes, emphasize it in the training section. In addition, if there are more research questions that the team wants participants to answer, please list them in the training section.

2. Line 150. The study mentioned that the team also collected the PHQ-9 depression scores. It will also be a good idea to report these data in the results. If you decide not to report the score, you do not need to mention it in training.

3. Line 248. The previous section mentioned that there are 14 participants; the total number of photos and narratives should be 70 if each participant presented five photos.

4. Line 253. The first theme, Mental Health and Substance Use Challenges, does not match the figures 1-3. A topic such as housing instability or the negative impacts of housing instability should fit figures 1-3 better.

5. Line 268-272: The participants’ quotations are not clear in the result section. Using separate paragraphs, quotation marks, or different fonts will be helpful for reading. For example, this passage has the same font and in the same paragraph and no quotation marks.

6. Line 301. The study should include a participant’s quotation for the statement about criminalizing persons with alcohol/substance problems.

7. Line 331: If participants gave every photo their narrative, the study results should at least include part of the narrative of the photo listed in the results. Figure 7 is a good example of having the photo and the narrative.

8. Figures 12-14 still belong to the results section. This is because they’re part of the findings.

9. The phrase “lastly” is overused in the article. Please replace it with other words.

Reviewer 2 Report

Homelessness is a severe problem, which is also very difficult to research.

So topically study has high merits. Also the new photovoice method seemed very interesting. However, maybe some more contextualization would be needed., Though the goal is to address and represent experienced lived problems and also empower the participants, it turned out that study is based on "socio-ecological" that is taught to participants. This is of course fine as such, but anyhow the outcome is that - in anthropological terms, emic is replaced with etic. The participants' voice is directed by reseachers' categories and ideas. Again, this is not wrong as such, but would definitely merit more thorough attention and reflection. That is, first of all, the authors could ópen up their 'socio-ecological' model, also justify why they think they are allowed impose their views on vulnerable subjects. - So this is also an ethical issue. Moreover, reflection could amount to considering both what this etic perspective gives, and what is lost when researchers have turned away from (more) emic perspective. The same is goes to selection of photos, which is also guided.

Also the way participants were guided to take and choose pictures merits more discussion. Clearly participants had chosen somewhat less controversical aspects of their life. This anyhow would need opening up and discussion on what it means for the representation of their life.

The number of participants is not  accounted clearly. In some place, 14 is mentioned, in some other places the number seems bit higher. Apparently, there have been various degrees of participation. Open this up.

Percentages are used for very small total number, e.g., 14. better would be to use fractions for small total numbers (definitely for smaller than 20).

Acronymns should be opened in the first use.

Also theoretical reflection on concepts like 'coping mechanism' would benefit from further layer of reflection.

So, the study does have a potential, but methodologically it is thinly framed.

It would merit from further theoretical work and ethical considerations.

Reviewer 3 Report

My only suggestion to improve the article is to explain why LGTBIq suffers from vulnerability. They are homeless people as well as black people but what about "white people" or heterosexual collective.

In my recommendation, I am asking the author to explain in four or five lines the origin of LGTBIG by reading the recommendation of the article. We explain the origin of sexual repression, the homophobia of the USA and some other countries all around the world. The confusion between neoliberalism and the patriarchal organization of the society is at the base of refusing LGTIQ people as well as the right-hand political parties against what they call gender ideology. This is exactly what this article needs.

Please read for a better explanation of LGTIQ explanations. Because they should explain why homophobia, lesbopbobia and ,in general, LGTBIQ collective is vulnerable. 
